# *Raffaelea quercina* sp. nov. Associated with Cork Oak (*Quercus suber* L.) Decline in Portugal

**Maria L. Inácio** [1,*] , **José Marcelino** [2] , **Arlindo Lima** [3,4] , **Edmundo Sousa** [1] **and Filomena Nóbrega** [1]

1 Instituto Nacional de Investigação Agrária e Veterinária, Quinta do Marquês, 2780-159 Oeiras, Portugal; edmundo.sousa@iniav.pt (E.S.); filomena.nobrega@iniav.pt (F.N.)
2 Entomology and Nematology Department, University of Florida, 1881 Natural Area Dr., Gainesville, FL 32608, USA; jmar06@gmail.com
3 LEAF—Linking Environment Agriculture and Food, Instituto Superior de Agronomia (ISA), University of Lisbon, Tapada da Ajuda, 1349-017 Lisboa, Portugal; arlindolima@isa.ulisboa.pt
4 LPVVA—Laboratório de Patologia Vegetal "Veríssimo de Almeida", Instituto Superior de Agronomia (ISA), University of Lisbon, Tapada da Ajuda, 1349-017 Lisboa, Portugal
* Correspondence: lurdes.inacio@iniav.pt

**Abstract:** Research Highlights: *Raffaelea quercina* sp. nov. is an ophiostomatoid fungus isolated from the ambrosia beetle *Platypus cylindrus*. The species occurs in symptomatic Portuguese cork oak trees, (*Quercus suber* L.), exhibiting vegetative decline. Background and Objectives: *Quercus suber* L. is a species restricted to the Mediterranean basin, of special economic importance as it constitutes the crucial raw material for the cork production industry, in particular for Portugal, the world's leading producer. Over the last three decades a progressive and alarming decline of cork oak trees has been observed across its distribution area, including Portugal. The ambrosia beetle *Platypus cylindrus*, commonly known as the oak pinhole borer, establishes symbiotic relationships with fungi from which it depends for survival and for oak colonization. Some of these fungi are ophiostomatoid species of the *Raffaelea* genus, known as ambrosia fungi associated with ambrosia beetles. Some *Raffaelea* species exhibit phytopathogenic activity causing wilting and/or death of trees. The objective of the present study is to identify the association between *P. cylindrus* and *Raffaelea* species in Portuguese cork oak stands showing symptoms of disease and decline. Materials and Methods: A total of 300 adult insects were collected as they emerged from cork oak logs, sampled from symptomatic trees. Axenic isolates of *Raffaelea* species were obtained from the beetles and their galleries in the trunks and identified based on morphological features and molecular analysis of the SSU and LSU rDNA regions. Results: Two *Raffaelea* species were identified, i.e., *R. montetyi* and a novel *Raffaelea* species closely related to *R. canadensis*. The novel species is morphologically and genetically characterized in this study, and erected as *Raffaelea quercina* M.L. Inácio, E. Sousa & F. Nóbrega, sp. nov. *Raffaelea quercina* constitutes a new phytopathogenic fungal species associated with *P. cylindrus* and cork oak trees exhibiting symptoms of vegetative decline. Conclusions: *Raffaelea* species appear to have a significant role in cork oak decline. Future research on the association between *P. cylindrus* and *Raffaelea* species, encompassing the trans-European and North African wide-range of cork oak stands, would further clarify the relationships between ambrosia beetles, associated fungi and cork oak decline, contributing to a better understanding of the phenomena and for strategies aiming to halt the continuous decline of the unique cork oak stands enclosed in the Mediterranean basin.

**Keywords:** ambrosia beetle; *Raffaelea* fungi; morphology; molecular phylogenetics; Mediterranean forest

## 1. Introduction

The cork oak (*Quercus suber* L.) is an oak species native to the Mediterranean zones of southwest Europe and northwest Africa. In Portugal, cork oak landscapes (*montados*) cover approximately 720 thousand ha [1], providing a vital source of income, especially due to the ability to produce cork, a raw material widely used for wine bottle stoppers.

*Montados* are listed under the EU Habitats Directive and are key protected habitats part of the Natura 2000 network and considered a EUNIS habitat type.

Since the 1890s, a cork oak decline has been reported in southwestern Portugal [2]. The occurrence of the decline was assumed as a complex phenomenon caused by the interaction of multiple biotic and abiotic factors [3], such as climatic variations (drought and frost), excessive harvesting and pruning, and attacks of insect pests and pathogens [4–7]. Coinciding with the gradual decline of cork oak stands increasing attacks by the wood borer *Platypus cylindrus* Fab. were reported [8–10]. The adults dig galleries in the wood of the host plants (*Q. suber*) while inoculating fungal ectosymbionts whose mycelium is fed on by both adults and larvae (Figure 1).

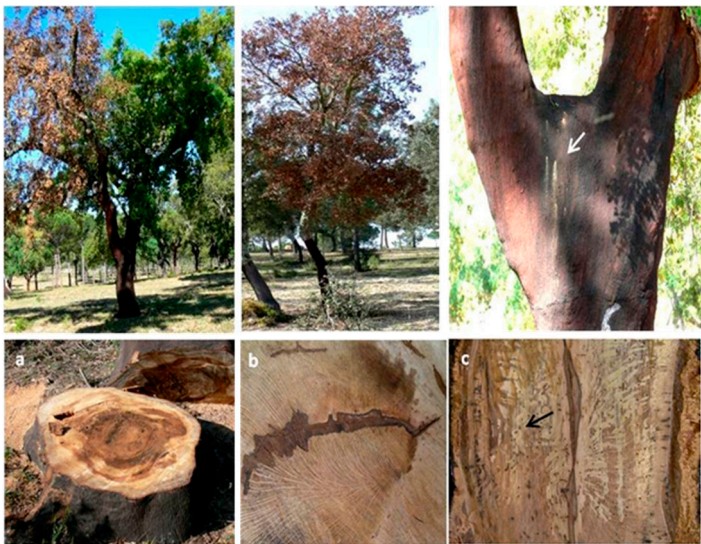

**Figure 1.** Symptoms on *Quercus suber* after the attack of *Platypus cylindrus*; white arrowed, sign of the presence of *P. cylindrus* on the trunk: orange sawdust coming out of the entry holes. Internal lesions caused by *P. cylindrus* in the trunk of *Q. suber* trees: (**a**) wood staining accompanying the galleries system (**b**) detail of a gallery section with surrounding wood staining brown (**c**) bore holes in cork oak heartwood in cross-section (black arrow).

In literature, more than 300 species of fungi are reported on cork oak, of which at least 100 are pathogenic [11–19]. Despite the numerous investigations carried out so far, it is difficult at first to assess which fungi could be phytopathogenic and thus responsible for participating in cork oak decline. However, several *Raffaelea* species were noticed provoking tree mortality and should be taking into consideration. *Raffaelea* (Arx & Hennebert 1965) is a genus of primarily asexual fungi including more than 20 species in Ophiostomatales [20,21] although more recently a species was found to have a sexual stage [22]. Some species in the genus, such as the causal agents of oak wilt in Japan (*R. quercivora*) [23] and Korea (*R. quercus-mogolicae*) [24], and laurel wilt in the USA (*R. lauricola*) [25] are important tree-killing pathogens.

The genus *Raffaelea* (*Ophiostomatales*) belongs to the ophiostomatoid fungi communities generally living in symbiosis with wood-boring bark beetles and their host's species. The symbiotic *Raffaelea* species exhibit frequent associations with the ambrosia beetle *Platypus cylindrus*, living in specialized structures called mycangia, particularly of female adults (Figure 2). *Platypus cylindrus* act as a vector for the dispersal of phytopathogenic fungi and this symbiotic behavior has been contributing for increasing the cork oak decline.

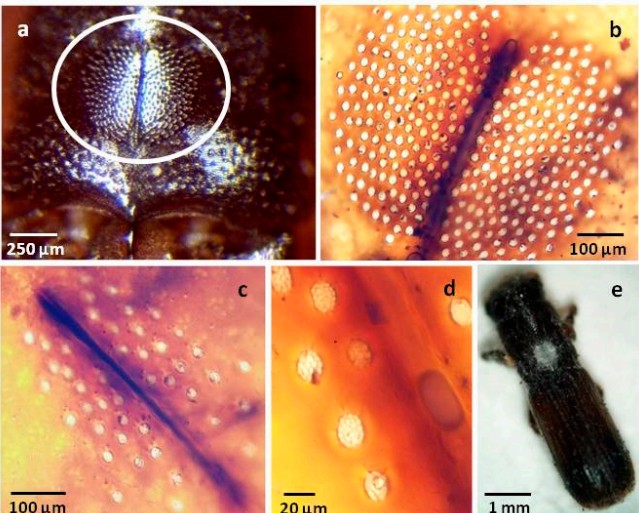

**Figure 2.** Mycangia of *Platypus cylindrus*: (**a**) female mycangia, (**b**) micrograph of female mycangia, (**c**,**d**) micrographs of male mycangia, (**d**) detail of integumentary pits filled with tightly packed spores of ambrosia fungi, (**e**) dying insect covered by the ambrosial mycelium coming out of the mycangia.

Species of *Raffaelea* are difficult to distinguish as they lack clearly defining morphological features, so identification can be based on rDNA sequences [26,27], in particular on the nuclear large subunit ribosomal DNA (LSU rDNA) [20].

The aim of this work was to ascertain the transportation and inoculation of *Raffaelea* species by *P. cylindrus* and to discuss their role in cork oak decline. During our research, adult insects emerged from symptomatic cork oak trees were collected and *Raffaelea*-like cultures were isolated, both from the insects and the galleries, and their identity determined based on morphological characters and DNA phylogenetic inference.

## 2. Materials and Methods

### 2.1. Beetle Collection and Fungal Isolation

From May 2005–2007, independent surveys were undertaken in two declining cork oak regions of Portugal, Ribatejo and Alentejo provinces. From cork oak trees severely infested by *P. cylindrus* and based on visible symptoms of decline, a total of 12 logs with ca. 30 cm diameter and 0.5 m length were collected, one from each tree. In the INIAV laboratories, three hundred adult *P. cylindrus*, 158 females and 142 males, emerged and were gathered live in fine mesh nets attached to a silicone joint on logs. Each beetle was dissected aseptically with iris scissors to obtain their mycangia, aiming to determine the fungal presence within these morphological structures. At the same time, pieces of wood containing fragments of the respective gallery were collected. All biological material was surface sterilized for 1 min in 1% sodium hypochlorite and then rinsed in sterile dH$_2$O and plated in malt extract agar (10 g MEA, Difco™, Franklin Lakes, NJ, USA, 15 g agar) amended with 500 mg/L cycloheximide and 500 mg/L streptomycin. Plates were incubated at room temperature in the dark for 1–2 weeks depending on fungal growth. Colonies of different size, color and mycelial pattern (concentric rings, yeasty growth, hyaline margins, etc.) were considered putative species.

Axenic cultures of each fungal strain were obtained (PC acronyms) and voucher specimens of each mycelial phenotype were deposited in the culture collection of INIAV Institute (Micoteca da Estação Agronómica Nacional (MEAN). In addition, mycelial plugs of at least three isolates of each putative fungal species were placed in the voucher specimen collection at Iowa State University (C acronyms) for storage and subsequent DNA extraction and sequencing (Table 1).

**Table 1.** Details of *Raffaelea* isolates obtained in this study (bold) and of strains representing species of *Raffaelea* retrieved from GenBank and used in phylogenetic analyses.

| Species | Collection No. [a] | Country | Host/Source | GenBank Accession Nrs. [b] | |
|---|---|---|---|---|---|
| | | | | SSU sDNA | LSU rDNA |
| *R. quercina quercina* sp. nov. | MEAN 1290 = PC05.007 = C2512 = CBS 147554 | Portugal (Santarém) | *Platypus cylindrus* | JF909509 | JF909537 |
| *R. quercina quercina* sp. nov. | MEAN 1291 = PC05.041 = C2516 | Portugal (Santarém) | *P. cylindrus* | JF909510 | JF909538 |
| *R. quercina quercina* sp. nov. | MEAN 1292 = PC05.042 = C2513 = CBS 147555 | Portugal (Santarém) | *Quercus suber, P. cylindrus* | JF909511 | JF909539 |
| *R. albimanens* | C2223 = CBS 271.70 | S. Africa | *P. externedentatus* | EU170269 | |
| *R. albimanens* | CBS271.70 | S. Africa | | | EU984296 |
| *R. ambrosiae* | C2225 = CBS 185.64 | UK | *P. cylindrus* | EU170278 | EU984297 |
| *R. amasae* | TUB F4290 | Germany | *Amasa* aff. *glaber* | AY858661 | |
| *R. arxii* | C2218 = CBS 273.70 | South Africa | *Xyleborus torquatus* | EU170279 | EU984298 |
| *R. brunnea* | C2229 = CBS378.68 | USA | *Monarthrum* sp. | EU170280 | EU177457 |
| *R. canadensis* | C2233 = CBS 168.66 | Canada | *P. wilsonni* | EU170270 | EU177458 |
| *R. canadensis* | CBS 168.66 | Canada | *P. wilsonni* | AY858665 | |
| *A. sulcati* | C592 = CBS 805.70 | Canada | *Gnathotrichus sulcatus* | EU170281 | |
| *R. ellipticospora* | C2346 | USA | *X. glabratus* | | EU177444 |
| *R. fusca* | C2254 | USA | *X. glabratus* | | EU177447 |
| *R. gnathotrichi* | C2219 = CBS 379.68 | USA | *G. retusus* | EU170282 | EU177460 |
| *R. lauricola* | C2203 | USA | *X. glabratus* | EU123076 | EU123077 |
| *R. lauricola* | C2339 = CBS 121567 | USA | *X. glabratus* | | EU177440 |
| *R. montetyi* | PC06.001, C2506 | Portugal (Alentejo) | *P. cylindrus* | JF909512 | JF909540 |
| *R. montetyi* | PC07.003, C2505 | Portugal (Alentejo) | *Quercus suber, P. cylindrus* | JF909513 | JF909541 |
| *R. montetyi* | PC06.038, C2504 | Portugal (Alentejo) | *P. cylindrus* | JF909514 | JF909542 |
| *R. montetyi* | PC06.039, C2514 | Portugal (Alentejo) | *Quercus suber, P. cylindrus* | JF909515 | JF909543 |
| **Species** | **Collection No. [a]** | **Country** | **Host/Source** | **GenBank accession nrs. [b]** | |
| | | | | SSU sDNA | LSU rDNA |
| | C2220 = CBS451.94 | Portugal (Alentejo) | *P. cylindrus* | - | EU177461 |
| | CBS451.94 | Portugal | *P. cylindrus* | | EU984301 |
| | C2221 = CBS463.94 | France | *P. cylindrus* | EU170283 | |
| | TUB F4264 | Germany | *X. dryographus* | AY497522 | |
| *R. quercivora* | PC10.919, MAFF919 | Japan | *P. quercivorus* | JF909517 | JF909547 |
| | PC10.921, MAFF921 | Japan | *P. quercivorus* | JF909516 | JF909546 |
| | JCM 15683 | Japan | *P. quercivorus* | | AB552937 |
| | MAFF410918 | Japan | *Q. crispula/P. quercivorus* | AB496428 | AB496454 |

**Table 1.** *Cont.*

| Species | Collection No. [a] | Country | Host/Source | GenBank accession nrs. [b] | |
|---|---|---|---|---|---|
| | | | | SSU sDNA | LSU rDNA |
| *R. santoroi* | CBS 399.67 | Argentina | *M. mutatus* | EU984261 | EU984302 |
| *R. subalba* | C2368 | USA | *X. glabratus* | | EU177441 |
| *R. subfusca* | C2253 | USA | *X. glabratus* | EU170268 | |
| *R. sulcati* | C2234 = CBS806.70 | Canada | *G. sulcatus* | EU170271 | EU177462 |
| *R. sulphurea* | C593 = CBS380.68 | USA | *X. saxeseni* | EU170272 | EU177463 |
| *R. tritirachium* | C2222 = CBS 726.69 | USA | *M. mali* | EU170273 | EU984303 |
| *Raffaelea* sp. | C1941 | USA | unknown | EU170274 | |
| *Raffaelea* sp. | C2224 = CBS326.70 | S. Africa | *P. externedentatus* | | EU177467 |
| *R. amasae* | CBS 116694 | Taiwan | *A. concitatus* | AY858660 | |
| *R. brunnea* | CBS378.68 | unknown | *Monarthrum* sp. | AY858654 | |
| *R. lauricola* | C2204 | USA | *X. glabratus* | EU170266 | |
| *R. subfusca* | C2335 = CBS121571 | USA | *X. glabratus* | | EU177450 |

[a] **PC**—fungal strains obtained from *Platypus cylindrus* and its galleries on *Quercus suber*, **MEAN** culture collection of INIAV Institute, Oeiras, Portugal; **C**, Iowa State University, Dept. of Plant Pathology, USA; **CBS**, Culture collection of the Westerdijk Fungal Biodiversity Institute, the Netherlands; **MAFF**, Mie University, Dep. Forest Pathology and Mycology, Japan. [b] Accession numbers of sequences newly produced (bold). SSU rDNA: small subunit region of of the ribosomal RNA gene; LSU rDNA: large subunit region of of the ribosomal RNA gene.

### 2.2. Fungal Morphology

Colony morphology was observed after 10 days of cultivation on 1.5% MEA medium at 25 °C in the dark. Colonized agar plugs (5 mm diam) were excised from actively growing 1 wk old cultures of three different isolates of each putative species. These discs were transferred to the centers of fresh dishes containing 20 mL 1.5% MEA. Growth rates were determined at temperatures ranging from 5–35 °C, at five-degree intervals, three and ten days after inoculation, in the dark. Colony diameter of six replicates cultures were calculated by averaging the twelve measurements. Mycelial colors were described using the terminology from Saccardo [28]. Tolerance to cycloheximide was assessed by measuring fungal growth on MEA amended with 100, 500 and 1.000 ppm cycloheximide after autoclaving. For fungal morphological characterization 3–5-day-old slide cultures [29] mounted in lactophenol were examined with light microscopy with differential interference contrast microscopy (BX-41 with DP11, Olympus, Center Valley, PA, USA). Fifty measurements were obtained for each taxonomically informative structure. For scanning electron microscopy (SEM), small wood blocks (5 × 2 × 5 mm) bearing fungal structures were fixed [30,31]. After fixation, samples were critical point dried, sputter coated twice with gold palladium (98:2) and examined using a JEOL 35 scanning electron microscope (JEOL, Peabody, MA, USA).

### 2.3. DNA Extraction, Amplification and Sequencing

Fourteen axenic fungal strains representing all the putative fungi species retrieved from *P. cylindrus* were selected for molecular analyses. Genomic DNA was extracted from mycelium plugs scraped with a sterile scalpel from the surface of individual cultures using the extraction kit Puregene® DNA Purification (Gentra Systems Inc., Minneapolis, MN, USA) following the manufacturer's instructions. Cultures of *Raffaelea montetyi* Morelet from Centraalbureau voor Schimmelcultures (strain CBS 451.94) and *R. quercivora* Kubono & Shin (isolates MAFF919 and MAFF921) from Mie University, Dep. Forest Pathology and Mycology, Japan, were used as reference material.

Polymerase Chain Reactions (PCR) were performed using PCR primer pairs NS1, NS3, NS4 and NS6 [32] for generating amplicons of the small subunit region of the ribosomal RNA gene (SSU rDNA) and the primers NL1 and NL4 [27] and LROR and LR5 to amplify the D1/D2 region of the large subunit region (LSU rDNA). All PCR reactions were performed in a 50 μL reaction containing 50–100 ng DNA template, 0.2 units of Taq Dream DNA polymerase (MBI Fermentas, Vilnius, Lithuania), 1× reaction buffer, 0.2 mM dNTPs, 4% (*v/v*) DMSO and 0.4 μM of each primer. PCR reactions were performed in a Biometra TGradient thermo cycler (Biometra, Göttingen, Germany) and the thermal cycling parameters included an initial denaturation at 95 °C for 5 min, 35 cycles at 94 °C for 1 min, annealing at 50 °C for SSU and 40 cycles at 55 °C for the LSU, and primer elongation at 72 °C for 1.5 min. PCR products were purified with QIAquick PCR Purification kit (Qiagen Inc., Valencia, CA, USA) following the manufacturer's instructions and sequenced in both directions at STABVida Sequencing Laboratory, Caparica (Portugal) on an ABI PRISM 3730xl DNA analyser (Applied Biosystems, Foster City, CA USA) using the same primers as those used for the amplification reaction for the SSU region and the primers LROR and LR3 for the LSU region [20]. Chromatograms were edited and consensus sequences were generated with Sequencher™ (Gene Codes Corp., Ann Arbor, MI, USA). Sequences were analyzed with all closely related sequences obtained from GenBank using BLAST (i.e., Maximum Identity of database segments against the subject sequence above 97%) and previously reported sequences [20,26,27,31]. Sequences were aligned individually with the Clustal W [33] and retrieved with Jalview software [34]. When necessary, subsequent manual adjustments were made. Sequences generated in this study were deposited in GenBank (accession numbers are included in Table 1). Primers were excluded from published sequences and sequence alignments.

### 2.4. Phylogenetic Analysis

Additional sequences using the BLAST algorithm on the National Center for Biotechnology Information (NCBI, https://blast.ncbi.nlm.nih.gov/ accessed on 29 June 2020) database were selected and incorporated in the analyses (Table 1). Phylogenetic analysis was conducted separately for the two rDNA regions (nSSU and nLSU) and the outgroup sequences were selected based on their genetic distance to Ophiostomatales used in the phylogenetic analyses [27,35,36]. Phylogenetic relationships were performed using MEGA X [37] and the Maximum Likelihood (ML) method based on the Kimura 2-parameter model. The robustness of ML tree was inferred using 1.000 bootstrap replicates.

## 3. Results

### 3.1. Fungal Isolation and Identification

Fungal isolations were performed directly from adult *P. cylindrus* emerged from symptomatic cork oaks, and from pieces of their galleries. From these beetles, 249 (83%) yielded a least one species of Ophiostomatales and only 4% yielded more than one species. It was possible to retrieve at least one species from each tree. A total of 270 individual ophiostomatalean isolates were obtained which, utilizing cultural features, microscopic characteristics and growth rate, were grouped according to colony morphologies. Among other fungi, a total of 14 *Raffaelea*-like colonies were selected for further studies.

### 3.2. Phylogenetic Analyses

Molecular identification of the 14 isolates based on SSU and LSU rDNA sequences confirmed that obtained sequences were species of *Raffaelea* (i.e., *Raffaelea* sp. and *R. montetyi*). The two species isolated were distinguished from each other by analysis of sequences and Maximum Likelihood (ML) phylogenetic analyses (Figures 3 and 4). A total of 1.568 base pairs were sequenced and the ML analysis revealed that one of the species represent a new species of *Raffaelea* belonging to the *Raffaelea sensu stricto* De Beer & Wingfield (2013) and described here as *R. quercina* sp. nov.

### 3.3. Morphology and Taxonomy

*Raffaelea quercina* M.L. Inácio, E. Sousa & F. Nóbrega, sp. nov. (Figure 5).

(1)   MycoBank: 838901
(2)   Holotype: LISE 96317
(3)   Etymology: Named after the host genus from which it was isolated, *Quercus*
(4)   Host/Distribution: On galleries of *Quercus suber* in Portugal = on mycangia of *Playpus cylindrus*

Description: Colonies effuse, yeast-like, with aerial brownish green floccose mycelium in the center, attaining a diam of 45–46 mm after 10 days on MEA, at 25 °C. Colony color from cream (27) to fuliginous (11) (Figure 5A). Hyphae hyaline and septate repeatedly branched and interlocked, hyphal ends sometimes developing into torulose swellings. Conidiophores unbranched, hyaline, solitary or clustered together to form hyaline sporodoquia. Conidiogenous cells without conspicuous scars from conidial dehiscence. Conidia blastosporic (sprout cells), unicellular and hyaline, usually solitary but sometimes in monilioid chains after germination *in situ*, smooth-walled, triangular to ovoid or fusiform, (6.7-)7.7-8.4(-10.9) × (2.5-)3.2-3.6(-5.0) μm.

Material examined: PORTUGAL, Chamusca (Santarém), in galleries of the insect *Platypus cylindrus* on declining *Quercus suber*, Maria L. Inácio, July 2005 (LISE 96,317 holotype; ex-type culture PC05.042 = MEAN 1292 = C2513 = CBS 147555); PORTUGAL, Chamusca (Santarém) in the mycangia of *Platypus cylindrus* emerged from *Quercus suber*, Maria L. Inácio, July 2005 (living culture, PC05.041 = MEAN 1291 = C2516); PORTUGAL, Chamusca (Santarém) in the mycangia of *Platypus cylindrus* emerged from *Quercus suber*, Maria L. Inácio, May 2005 (living culture, PC05.007 = MEAN 1290 = C2516 = CBS 147554).

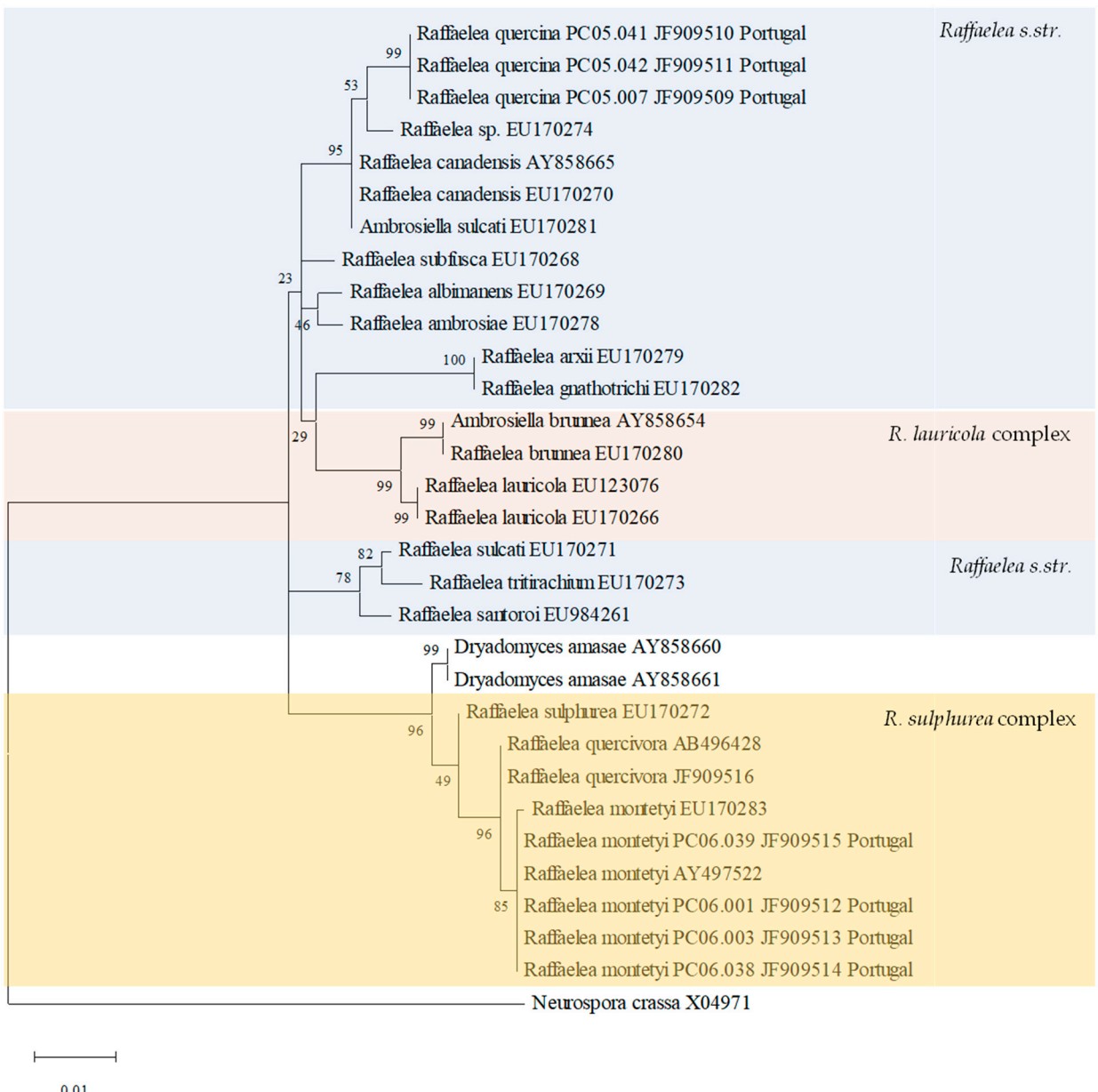

**Figure 3.** Phylogenetic relationships of *Raffaelea* isolates collected from *Platypus cylindrus* emerged from cork oak logs based on the sequence alignment of the partial sequence (1.028 bp) of the SSU region or the rDNA. The phylogram was generated by ML analyses with 1.000 bootstrap replications. Bootstrap values are indicated at the nodes. The analysis involved 31 nucleotide sequences. There were a total of 1.022 positions in the final dataset. Evolutionary analyses were conducted in MEGA X [37].

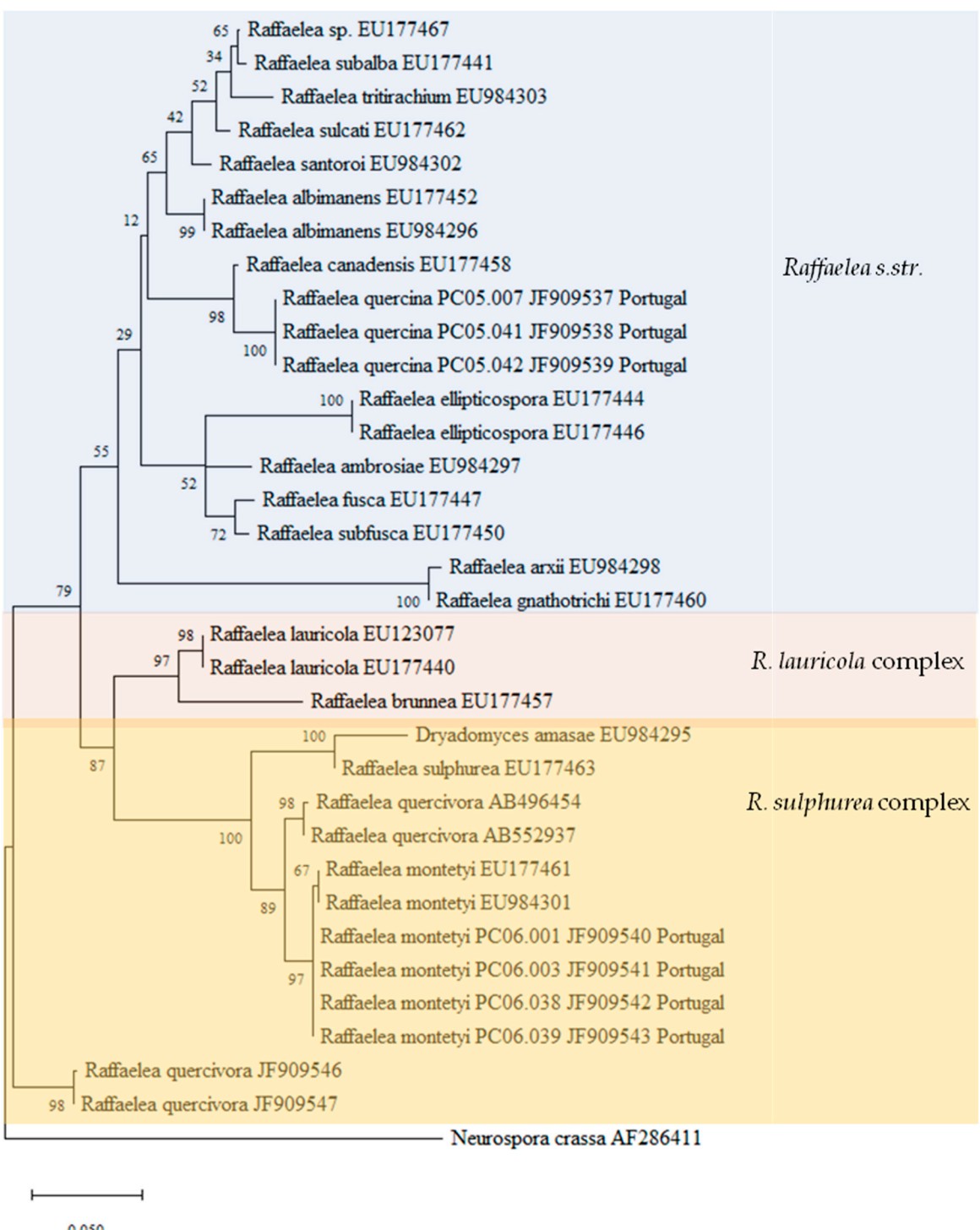

**Figure 4.** Phylogenetic relationships of *Raffaelea* isolates collected from *Platypus cylindrus* and its galleries on cork oak logs based on the sequence alignment of the partial sequence of the LSU region or the rDNA. The phylogram was generated by ML analyses with 1.000 bootstrap replications. Bootstrap values are indicated at the nodes. The analysis involved 34 nucleotide sequences. There were a total of 540 positions in the final dataset. Evolutionary analyses were conducted in MEGA X [37].

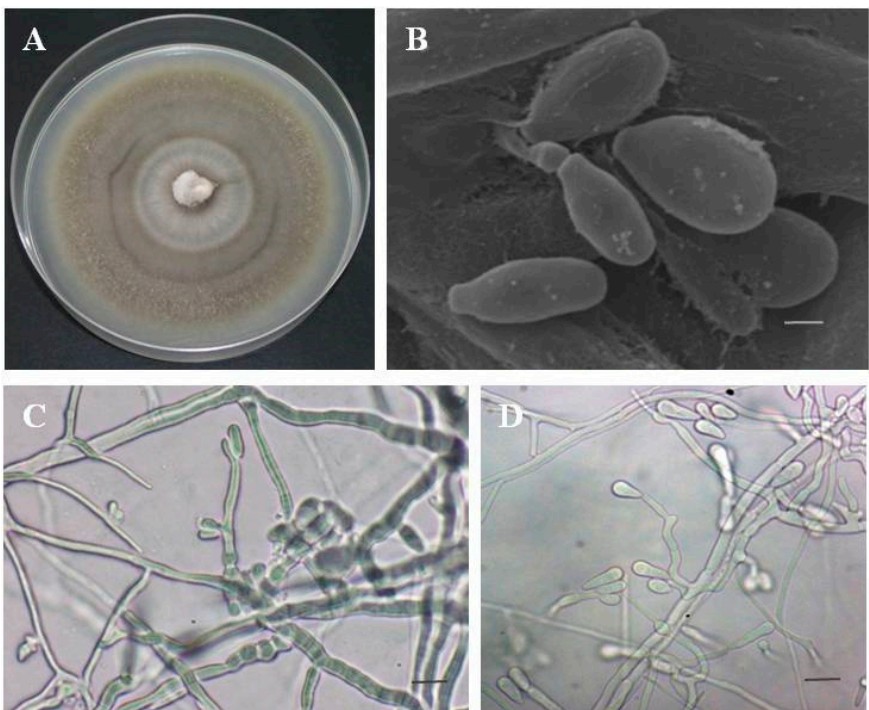

**Figure 5.** Culture, conidiophores and conidia of *Raffaelea quercina* sp. nov. (isolate PC05.042 = MEAN 1292) (**A**) Colony morphology after 2 weeks on malt extract agar in a 90 mm diameter plate. (**B**) Scanning electron micrograph showing ovoid conidia (bar = 1 μm) (**C,D**). Light micrographs of conidia and conidiogenous cells without conspicuous scars from conidial dehiscence (bar = 10 μm).

Notes: *Raffaelea quercina* is morphologically similar to *R. canadensis* but colonies never wrinkled at the centre and with no cracks in agar under the wrinkled area; odour of ripened bananas never present as in *R. canadensis* cultures; *R. canadensis* spores usually globose (5-)6.5-8.0(-13) μm while triangular to ovoid or fusiform in *R. quercina* (6.7-)7.7-8.4(-10.9) × (2.5-)3.2-3.6(-5.0) μm. They are clearly separated in the phylograms of both SSU and LSU rDNA regions (see Figures 3 and 4).

The new Portuguese *Raffaelea* species presents a stable position in the inferred phylogenetic relationships. The species is clearly separated in the phylogram based on SSU and LSU sequence data, *R. quercina* isolates formed a separate clade with strong support values on the analyses performed.

## 4. Discussion

In this research, *Raffaelea quercina* is described as a new species of the genus *Raffaelea*, isolated from the ambrosia beetle *Platypus cylindrus* and its galleries on symptomatic cork oak trees in Portugal in 2005 and in the following years during entomological surveys (data not shown).

The known association between *Raffaelea* species and *P. cylindrus* implies that *R. quercina* was probably transmitted to the trees by the insects. Based on the morphology and molecular phylogenetic analyses of SSU and LSU sequence data, this taxon proved to be distinct from other species from the genus.

*Raffaelea montetyi* was the predominant *Raffaelea* species isolated from *P. cylindrus* in Portugal [4], consistent with the results already described [26,38]. Isolations from adults yielded *R. montetyi* in about 45% of the asexual cycloheximide-tolerant symbionts, being *R. quercina* less frequent (33 individual isolates corresponding to 12.2% of the isolates). In contrast to the slow growth of most of ambrosia fungi, *R. montetyi* grows rapidly being able to colonize the galleries system much faster than the other symbionts. Taking into consideration previous studies on *P. cylindrus* mycoflora [12,39–41] where *R. montetyi* was not identified, its presence on the symbiosis appears to constitute an ecological adaptation

aiming faster and efficient host colonization. *Raffaelea montetyi* had previously been isolated from *P. cylindrus* [38] but confirmation of its implication with declining oaks was not assessed until the work of Inácio et al. [4].

Portuguese *Raffaelea* near *R. canadensis* isolates retrieved from *P. cylindrus* and its galleries in cork oak were identical to the holotype *R. canadensis*, both morphologically and genetically, with modest differences. Pairwise *p* distances between Portuguese and holotype strains available in GenBank showed nucleotide differences of ca. 2% (5 bp) for the more discriminate nuclear rDNA LSU regions. This far, *R. canadensis* had been solely identified in association *Gnathotrichus sulcatus* Lec. (as *A. sulcati*) in *Psedotsuga menziesii* (Douglas fir) [42]. However, a recent association of an unidentified *Raffaelea* species similar to *R. canadensis* (1.9% bp divergence for the LSU region, EU177458) was also reported from another ambrosia beetle, *Xyleborus glabratus* Eichh. in *Persea borbonia* (redbay) hosts [43]. Based on the comprehensive molecular and morphological data from our study we believe the new associations recently found with the ambrosia beetles *P. cylindrus* may consist of unreported *Raffaelea* closely related to *R. canadensis* with a selective preference for associations with a specific insect species. *Raffaelea quercina* is therefore newly described both in associations with *P. cylindrus* and with cork oak declining trees.

It has generally been accepted that one or few fungal species are associated with a particular ambrosia beetle species [42,44] but recent studies pointed out that the symbionts of ambrosia beetles are more diverse [45–47], more promiscuous [48] and more competitive than previously assumed and *Raffaelea* species may compete among each other for entrance to and growth within the mycangium [20,43]. Species of *Raffaelea* have been related with mass mortality of oak trees in Japan [23,27] and laurel wilt of redbay in the USA [49]. Portuguese *Raffaelea* isolates have a proven pathogenicity against cork oak seedlings and therefore could have a significant role in cork oak decline, in particular *R. montetyi* [4]. *Raffaelea quercina* was shown to cause wilting on the inoculated seedlings but with a low level of aggressiveness, as obtained for other *Raffaelea* species [50,51]. A comprehensive morphological and phytopathological assessment, coupled with a robust phylogenetic analysis, of the complex of fungi associated with *P. cylindrus* populations across a trans-European and north African wide-range number of cork oak stands could further clarify the relationships of ambrosia beetles and their associated fungi with cork oak decay and contribute to halt the spread of the decline in the unique cork oak landscapes enclosed in the Mediterranean basin.

## 5. Conclusions

A novel fungal species, *Raffaelea quercina* was described. Our previous works demonstrated that *R. quercina* is a pathogenic fungus towards *Q. suber* seedlings as *R. montetyi*, hence probably contributing to cork oak decline and to the establishment of its associated ambrosia insect, *Platypus cylindrus*.

Cork oak forests represent a very specific, delicately balanced ecosystem which only persists in the Mediterranean basin. It is therefore of major concern that over the last three decades an alarming decline of trees has increased across its distribution area, namely in the representative Portuguese cork oak stands. Being cork oak decline a multifactorial process, several causes have been pointed out as contributors to tree mortality and loss of vigour, namely biotic factors. *Platypus cylindrus* emerged as a determinant factor in the decline of stands and its population outbreaks in the last decades have caused heavy economic damages since cork loses its quality and ultimately trees death overcomes. The symptoms and signs exhibited by cork oaks attacked by *P. cylindrus*, including the presence of numerous entry holes and profuse sawdust emerging from these holes, do not reveal the real dimension of the attack intensity within the trunk. Only cutting the attacked tree and following the insect tunnels, is possible to ascertain the extent of the galleries into the sapwood. The galleries system often becomes quite complex. Coupled with this extensive boring activity, the inoculation of ambrosia fungi is part of the insect strategy to establish its offspring in the host trees.

In terms of role in oak decline, the combined action of *P. cylindrus* massive attacks and extensive boring with the inoculation of ambrosia fungi, leads to an increase of tree mortality enhanced in these past three decades through these new associations with more aggressive and wilt causing fungi. Understanding the ecology and population dynamics of *P. cylindrus*-associated fungi is important for the surveillance and management of the beetle-fungal complex, and could improve prediction and modeling. The ambrosia beetles and their associated fungi constitute a small part of a much larger food web, the complexities of which we have barely start to understand. There are indeed many unanswered questions about the extraordinary complexity between these wood-inhabiting beetles, the assembly of fungi which they transmit, and the tree which supports the whole community. We believe the research described herein is a contribution to clarify focal aspects of the pathology of cork oak decline, aiming to preserve the economic and cultural heritage of the unique cork oak stands and landscapes present in the Mediterranean.

**Author Contributions:** Conceptualization, M.L.I. and E.S.; methodology, M.L.I., E.S. and F.N.; software, F.N. and J.M.; validation, M.L.I., J.M., A.L., E.S. and F.N.; formal analysis, M.L.I., J.M., A.L., E.S. and F.N.; resources, M.L.I., E.S. and F.N.; data curation, M.L.I. and F.N.; writing—original draft preparation, M.L.I. and F.N.; writing—review and editing, M.L.I., J.M., A.L., E.S. and F.N.; visualization, M.L.I., J.M., A.L., E.S. and F.N.; supervision, E.S. and A.L.; project administration, M.L.I.; funding acquisition, M.L.I., E.S. and F.N. All authors have read and agreed to the published version of the manuscript.

**Funding:** This research was supported in part by a grant from the Fundação para a Ciência e a Tecnologia (FCT) BD/26033/2005 and by Instituto Nacional de Investigação Agrária, I.P.

**Institutional Review Board Statement:** Not applicable.

**Informed Consent Statement:** Not applicable.

**Data Availability Statement:** The data presented in this study are openly available in [GenBank].

**Acknowledgments:** The authors would like to thank Marina Cardoso for the technical assistance and Joana Henriques for the valuable contribution in some parts of the practical work, for the enthusiasm and passion for the ambrosia fungi shared with us for some years. We are also grateful to Thomas Harrington of Iowa State University (USA) and Wilhelm de Beer and. Michael Wingfield of FABI, University of Pretoria (South Africa) for teaching us so many important things about *Raffaelea* fungi, and for receiving our precious strains.

**Conflicts of Interest:** The authors declare no conflict of interest.

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
