# Peer review of "Raffaelea quercina sp. nov. Associated with Cork Oak (Quercus suber L.) Decline in Portugal"

_forests, doi:10.3390/f12040513_

Round 1
Reviewer 1 Report
Describe an unknown plant pathogen fungi is important. this paper does well in the fungal description but really miss several important data such as beta-tubulin gene and new recently described Raffaelea fungi. One more big deal is the title, Title should represent the story. but i could tell the paper is focus on the description of a new Raffaelea fungi. It did not include tree pathogenicity result to prove the decline. the current title is confusing me. so i would suggest the title should be "A new ophiostomatalelean fungi Raffaelea quercina sp. nov associate with ambrosi beetle Platypus cylindrus in Potugal" based on the main content are on isoaltion from beetle and beetle gallery. The author mentioned forest in the introduction and discussion only, so i am surprised why author did not submit it to a mycology journal because its main result is a new fungus species. not forest. I do agree this paper is important to beetle damage in forests. but author need to add more detail of that tree damage or tree survey result. so the content could fit the title.
line 23 "sp."should be "spp."
line 70 delete "so"
line 106 "nuclear large subunit ribosomal DNA (LSU rDNA)". no. It's incorrect. For species description. Most recently paper includes LSU and β-tubulin gene. see R. promiscua by Nel et al 2021, R. borbonica by Procter et al 2020, and five Raffaelea by Simon et al (2016). β-tubulin is very important in Raffaelea fungi. Only three Raffaelea species are missing β-tubulin now (R. gnathotrichi, R. scolytodis, R. rapaneae). Also, personally, i would also recommend add ITS genes if possible. ITS will help in BLAST results for most mycologists. in conclusion, add a phylogeny of β-tubulin is better but not compulsive. but add β-tubulin is really necessary.
line 116 line 222 How many tree were surveyed and cut into "12 logs"? please add more detail. Also it looks like author only sampled from at most 12 tree. It's really a very small sample number. so i doube it could support the title.
the same to result 3.1 is make me very disppointing. Author title the paper as fungi associate with the tree decline. however, i could not tell how it happen. only 12 log present in this paper. I don't know how many tree were surveyed by author? also in here. author only mentioned how many beetles own the fungi and the number of fungi species. but i could not tell how many beetle have R. quervine and how many have R. montetyi? if beetle emerged from all 12 log all with those Raffaelea fungi.
line 301 "stable position in the inferred phylogenetic relationships" which is in conflict to phylogeny of SSU. In SSU the branch is only support by "53" bootstrap values which is low and not well support. author should explain it or redo the phylogeny. SSU is not good to test the phylogeny of Raffaelea.
line 372 "distinct from other species". not really. some new published Raffaelea species are not included in this phylogeny. such as R. promiscua by Nel et al 2021, R. borbonica by Procter et al 2020
line 387 ""ca. 2% for" add how many loci different in the gene
line 390 [42] is incorrect. the correct is "Funk A. 1965. The symbiotic fungi of certain ambrosia beetles in British Columbia. Canad. J. Bot. 43: 929–932.
line 391 ""R. canadensis (1.9% bp divergence for the LSU region)" add association genebank number
line 404 add"in USA" after "redbay'.
line 416 "Our works" should be "Our previous works". because in this paper author did not present pathogenicity result
line 445 this last sentence is a statement but it's inappropriate in this paper, because this paper does not clarify focal aspects of the pathology of cork oak decline. as mentioned in above. the author only describes the fungi and few isolation date. the author should be cautious to be not overstate the result.
Table 1 There are many mistakes in Table1. Author should be carefully to cite those information. many of previous record are incorrect. please update. I only pointed some.
1. host of R. albimanens is "Crossotarsus externedentatus"
2. host of R. canadensis is "Euplatypus wilsonni"
3. country (Germany) and host (CBS..) of R. canadensis are incorrect
4. country of R. quercivora should be "Japan" which is more offical
5. host of R. santoroi is "Megaplatypus mutatus"
6. host of R. sulphurea is "Xyleborinus saxeseni". not a Xyleborus. so it's not "X."
7. country of R. amasae and R. brunnea is not "Germany". Please double check those origin.
8. If author add the column of "collection no.". It should also add the information of other Raffaelea. not only the new Raffaelea isolation from this study.
Figure 3 and Figure 4
the blue color should cover all Raffaelea s. str. not only the branch of new species. So many other Raffaelea s.str fungi are missing from the blue color. please change it.
Author Response
Dear Reviewers,
Thank you for the careful and thorough reading of our manuscript “Raffaelea quercina sp. nov. associated with cork oak (Quercus suber L.) decline in Portugal”
We appreciate the thoughtful comments and constructive suggestions, which help us to improve the quality of the document. However, we were a bit hampered with Reviewer #1 comments, as they were difficult to understand. Nevertheless, we have tried to reply to each of them and we are now convinced that our MS meets the requirements to be published in this Journal.
As we were asked, all revised items are marked along the text. Also, the answers to reviewers’ comments are written below. [original reviewers’ comments in bold)
Thank you once again for your advice and review.
The authors
Dear Reviewers,
Thank you for the careful and thorough reading of our manuscript “Raffaelea quercina sp. nov. associated with cork oak (Quercus suber L.) decline in Portugal”
We appreciate the thoughtful comments and constructive suggestions, which help us to improve the quality of the document. However, we were a bit hampered with Reviewer #1 comments, as they were difficult to understand. Nevertheless, we have tried to reply to each of them and we are now convinced that our MS meets the requirements to be published in this Journal.
As we were asked, all revised items are marked along the text. Also, the answers to reviewers’ comments are written below. [original reviewers’ comments in bold)
Thank you once again for your advice and review.
The authors,

Reviewer 2 Report
The current study focused on the investigation of associations between the ambrosia beetle, Platypus cylindrus, and Raffaelea quercina fungi described as potential causes of wilting and decline of cork oak stands. Throughout the work the authors identify multiple fungal species, and identify a novel species as well, which is the most novel aspect of this work. Better understanding of these associations and their effect on oak forests is a step forward in potentially finding avenues towards controlling infestations in the future, and aid conservation efforts.
Author Response
Dear Reviewers,
Thank you for the careful and thorough reading of our manuscript “Raffaelea quercina sp. nov. associated with cork oak (Quercus suber L.) decline in Portugal”
We appreciate the thoughtful comments and constructive suggestions, which help us to improve the quality of the document. However, we were a bit hampered with Reviewer #1 comments, as they were difficult to understand. Nevertheless, we have tried to reply to each of them and we are now convinced that our MS meets the requirements to be published in this Journal.
As we were asked, all revised items are marked along the text. Also, the answers to reviewers’ comments are written below. [original reviewers’ comments in bold)
Thank you once again for your advice and review.
The authors,
Reviewer 3 Report
In the manuscript, the authors describe a new species of fungi from Portugal, related to the decline of cork oak forests in western Mediterranean. The work is interesting, in line with the scope of the journal, and its scientific value is relevant for the conservation of coark oak forests. I only have some minor comments for its improvement. Furthermore, though the manuscript is well-written, I believe that the English language could benefit from a revision by an expert or native speaker.
Introduction: please add a statement that cork oak forests are protected in Europe, being both a Natura 2000 and a EUNIS habitat.
Lines 107-113: These are methods. Here, you should rather insert a brief statement about the aims of your work.
Table 1: this table anticipates the results if put in M&M. I suggest to move it in the Results section.
Lines 170 and 240: wrong way to cite, please correct.
Line 172: missing period at the end of the sentence.
Lines 226-228: this is a repetition of methods.
Line 232: Rafaelea-like.
Line 238 and elsewhere in the text: 1,568 instead of 1568.
Line 381: P. cylindrus in italics.
Line 415: "Our previous works"
Author Response
Dear Reviewers,
Thank you for the careful and thorough reading of our manuscript “Raffaelea quercina sp. nov. associated with cork oak (Quercus suber L.) decline in Portugal”
We appreciate the thoughtful comments and constructive suggestions, which help us to improve the quality of the document. However, we were a bit hampered with Reviewer #1 comments, as they were difficult to understand. Nevertheless, we have tried to reply to each of them and we are now convinced that our MS meets the requirements to be published in this Journal.
As we were asked, all revised items are marked along the text. Also, the answers to reviewers’ comments are written below. [original reviewers’ comments in bold)
Thank you once again for your advice and review.
The authors,
Response to reviewer #3 comments
Introduction: please add a statement that cork oak forests are protected in Europe, being both a Natura 2000 and a EUNIS habitat.
Thank you for this suggestion. We have added ”Montados are listed under the EU Habitats Directive and are key protected habitats part of the Natura 2000 network and considered an EUNIS habitat type.”
Lines 107-113: These are methods. Here, you should rather insert a brief statement about the aims of your work.
The Reviewer is right and we have altered this sentence to:
The aim of this work was to ascertain the transportation and inoculation of Raffaelea species by P. cylindrus and to discuss their role in cork oak decline. During our research, adult insects emerged from symptomatic cork oak trees were collected and Raffaelea-like cultures were isolated, both from the insects and the galleries, and their identity determined based on morphological characters and DNA phylogenetic inference.
Table 1: this table anticipates the results if put in M&M. I suggest to move it in the Results section.
We totally agree with this opinion but it is the standard procedure for this type of manuscripts where a table with all the isolates used for comparison integrates the M&M section, including the isolates to be identified.
Lines 170 and 240: wrong way to cite, please correct.
Done, thank you very much.
Line 172: missing period at the end of the sentence.
Done, thank you.
Lines 226-228: this is a repetition of methods.
The Reviewer is right and we have changed the sentence, leaving only the necessary for a better understanding.
Line 232: Rafaelea-like.
Done, thank you (only the fungal collection numbers are left as provided by the curators).
Line 238 and elsewhere in the text: 1,568 instead of 1568.
Done, thank you.
Line 381: P. cylindrus in italics.
Done, thank you.
Line 415: "Our previous works"
The reviewer is right! Done, thank you.
Round 2
Reviewer 1 Report
To author
there are still some suggestion from me.
- the title is "Raffaelea quercina sp. nov. associated with cork oak (Quercus suber L.) decline in Portugal". but only result3.1 (line 242-255) presents the result of "R. quercine associated". Author indicate the 270 fungal isolates are isolated from beetle and gallery on 12 decline tree. but we could not tell the result corresponding to the title. When the author mentioned "R. quercine associated". At least we should know how many tree in the 12 sample we could revive the R. quercine or how many beetle/gallery could revive the R. quercine? Without the data of frequency, I am not agree the title corresponding to the main data. I knew author group published several paper previously about this Raffaelea new species. but those previous data did not cited clearly in the result and discussion. i am still not satisfied with data of "associated".
- sorry to my previous statement "add a phylogeny of β-tubulin is better but not compulsive. but add β-tubulin is really necessary" makes confusing. I would like to highlight how important the data of β-tubulin. So if possible. i would highly recommend author add data of β-tubulin. ideally a phylogeny of β-tubulin. if that was difficult. add the sequence of β-tubulin is also good and necesary. As the author explained in Harrington (2010) and de Beer (2013) work. Previously(15 years ago), people did not know the importance of β-tubulin in this group. but recently more and more paper found β-tubulin is critical and all new Raffaelea and other ophiostomatalean fungi described in recently 5 years were included β-tubulin. For those strains missing β-tubulin data in previous work. People from FABI and other institutes and universities were trying to add those missing β-tubulin data in the new publication. so in here, i would say we follow the new "rule". If not, the reason should be mentioned, so that other mycologists could know what happens and include β-tubulin in their new work.
line 24-26. the main objective of this study is the description of new species. add those to the abstract.
line 54 "Montados" why it's italic
line 187 Not sure why author mentioned ITS primers here. Neither result or discussion. none of ITS sequences was included.
line 360 Author wrote "Portuguese Raffaelea near R. canadensis". I feel it's the R. quercina here. Why not say R. quercina directly? named it "Portuguese Raffaelea" is confused.
Table1
two "R. canadensis" are same sample based culure collection number. so they all from Canada, not Taiwan.
two "R. albimanens" are same sample based culure collection number. so they all from S. Africa, not Canada.
two "R. ambrosiae" are same sample based culture collection number. they should be one record.
If author could not tell those old incorrect data in the column of ‘country’ and ‘Host/Source’. I would suggest deleting those two columns.
Author Response
Dear Reviewer,
Thank you for the thorough reading of our manuscript “Raffaelea quercina sp. nov. associated with cork oak (Quercus suber L.) decline in Portugal”.
We appreciated your comments, some insistently made, which we consider a big effort from the reviewer’s side for us to improve our manuscript.
We have tried to reply to each of them and we are now convinced that our MS meets the requirements to be published in this Journal.
As we were asked, all revised items are marked along the text. Also, the answers to reviewer’s comments are written below. [original reviewer’s comments in bold)
Thank you once again for your advice and review.
The authors
Response to reviewer #1 comments
the title is "Raffaelea quercina sp. nov. associated with cork oak (Quercus suber L.) decline in Portugal". but only result3.1 (line 242-255) presents the result of "R. quercine associated". Author indicate the 270 fungal isolates are isolated from beetle and gallery on 12 decline tree. but we could not tell the result corresponding to the title. When the author mentioned "R. quercine associated". At least we should know how many tree in the 12 sample we could revive the R. quercine or how many beetle/gallery could revive the R. quercine? Without the data of frequency, I am not agree the title corresponding to the main data. I knew author group published several paper previously about this Raffaelea new species. but those previous data did not cited clearly in the result and discussion. i am still not satisfied with data of "associated".
As already mention, we consider the title suits completely the work done and presented in the manuscript. Furthermore, the work is totally within the scope of Forests – special issue “Insects as Vectors of Forest Diseases”.
Even if the findings are from some years ago, we are convinced it is a valuable contribution for the knowledge about the interactions between the insect P. cylindrus and its ambrosial symbionts in relation to cork oak stands decline and very much aligned with the international concerns on the subject.
The authors published several papers in which the results are detailed in terms of numbers and graphs and they are all cited in this manuscript. Thus, our goal was not to repeat published information but to provide new findings instead, there is, the discovery and characterization of a new Raffaelea species related to cork oak decline. Anyway, we provided enough information and added the requested sentence. This is evident in the paragraphs:
“Fungal isolations were performed directly from adult P. cylindrus emerged from symptomatic cork oaks, and from pieces of their galleries. From these beetles, 249 (83%) yielded a least one species of Ophiostomatales and only 4% yielded more than one species. It was possible to retrieve at least one species from each tree. A total of 270 individual ophiostomatalean isolates were obtained which, utilizing cultural features, microscopic characteristics and growth rate, were grouped according to colony morphologies. Among other fungi, a total of 14 Raffaelea-like colonies were selected for further studies.
“Raffaelea montetyi was the predominant Raffaelea species isolated from P. cylindrus in Portugal [4], consistent with the results already described [26,38]. Isolations from adults yielded R. montetyi in about 45% of the asexual cycloheximide-tolerant symbionts, being R. quercina less frequent (33 individual isolates corresponding to 12.2% of the isolates).
Portuguese Raffaelea isolates have a proven pathogenicity against cork oak seedlings and therefore could have a significant role in cork oak decline, in particular R. montetyi [4]. Raffaelea quercina was shown to cause wilting on the inoculated seedlings but with a low level of aggressiveness, as obtained for other Raffaelea species [50,51].”
Finally, we would like to emphasize that most of Raffaelea/Ophiostoma species descriptions in reference articles do not have numerical data associated (isolation frequency, number of host trees from which isolations were made, …) and yet they constitute very valuable reports.
sorry to my previous statement "add a phylogeny of β-tubulin is better but not compulsive. but add β-tubulin is really necessary" makes confusing. I would like to highlight how important the data of β-tubulin. So if possible. i would highly recommend author add data of β-tubulin. ideally a phylogeny of β-tubulin. if that was difficult. add the sequence of β-tubulin is also good and necesary. As the author explained in Harrington (2010) and de Beer (2013) work. Previously(15 years ago), people did not know the importance of β-tubulin in this group. but recently more and more paper found β-tubulin is critical and all new Raffaelea and other ophiostomatalean fungi described in recently 5 years were included β-tubulin. For those strains missing β-tubulin data in previous work. People from FABI and other institutes and universities were trying to add those missing β-tubulin data in the new publication. so in here, i would say we follow the new "rule". If not, the reason should be mentioned, so that other mycologists could know what happens and include β-tubulin in their new work.
Although we agree that β-tubulin is a useful protein coding gene to discriminate Raffaelea spp., this cannot be the sole driving force dictating species assignation. It is rather a supporting tool, as the genes we present in the paper. Adding β-tubulin would not change findings. It would just increase their support. Apart from β-tubulin many other recent novel approaches could have been used, e.g., whole genome sequencing and transcriptomic analyses, to determine species assignation and pathogenesis. See Zhang et al (2020). Genomic and transcriptomic insights into Raffaelea lauricola pathogenesis. BMC Genomics 21 https://doi.org/10.1186/s12864-020-06988-y.
To include a phylogeny of β-tubulin like the reviewer suggests, or more accurately a phylogeny of Raffaelea species based on β-tubulin, is not an addition to the paper but rather a reconstruction of its content. A content for which the remaining reviewers concur as solid for publication.
As reviewer 1 pointed out, many institutions and researches added β-tubulin to phylogenies, as new findings occur, and these were published in subsequent articles from these same authors. We intend to do the same as human and financial resources become available, however, it is not a reasonable request to redo an entire phylogeny, sequence new voucher specimens and reconstruct the present paper to add this gene. Other novel approaches as listed in Zhang et al (2020) are even more valuable than this gene, and we hope in future work to pursue both β-tubulin and the strategy listed in Zhang et al. (2020), to further enhance our knowledge on this genus, the novel species and its etiology and pathogenicity.
line 24-26. the main objective of this study is the description of new species. add those to the abstract.
In fact, that was never our main goal! Our objective was to determine the cause of the decline of cork oak trees affected by the woodborer insect Platypus cylindrus, investigating the species of obligatory fungi (not necessarily new species) involved in its establishment and host colonization.
line 54 "Montados" why it's italic
Because it is a Portuguese designation, not an English noun.
line 187 Not sure why author mentioned ITS primers here. Neither result or discussion. none of ITS sequences was included.
Thank you, you are correct and this part was erased.
line 360 Author wrote "Portuguese Raffaelea near R. canadensis". I feel it's the R. quercina here. Why not say R. quercina directly? named it "Portuguese Raffaelea" is confused.
We are trying to describe the process and rationale that led us to identify the species as new. This process initiated by finding similarities with R. canadensis which were posteriorly investigated, as described in the narrative, and ultimately led to the new identification. We need to be succinct, yes, but clearly document the procedure leading to the identification of a new species, which is not automatic based on resemblance. However we added Raffaelea quercina, thank you.
Table1
two "R. canadensis" are same sample based culure collection number. so they all from Canada, not Taiwan. Correction made, thank you.
two "R. albimanens" are same sample based culure collection number. so they all from S. Africa, not Canada. Correction made, thank you.
two "R. ambrosiae" are same sample based culture collection number. they should be one record. Correction made, thank you.
If author could not tell those old incorrect data in the column of ‘country’ and ‘Host/Source’. I would suggest deleting those two columns.
Thanks to Reviewer 1, we have been able to correct our original table resulting in a very detailed and complete Table. Thus, it would be a waste deleting some information since we consider it useful and valuable. As in other reference works, sometimes there are unknown data but still the information is given.
